# Effect of *ipt* Gene Induction in Transgenic Tobacco Plants on Hydraulic Conductance, Formation of Apoplastic Barriers and Aquaporin Activity under Heat Shock

**DOI:** 10.3390/ijms24129860

**Published:** 2023-06-07

**Authors:** Lidiya Vysotskaya, Guzel Akhiyarova, Oksana Seldimirova, Tatiana Nuzhnaya, Ilshat Galin, Ruslan Ivanov, Guzel Kudoyarova

**Affiliations:** Ufa Institute of Biology, Ufa Federal Research Center, Russian Academy of Sciences, Pr. Octyabrya 69, 450054 Ufa, Russiatanyawww89@mail.ru (T.N.);

**Keywords:** immunolocalization of cytokinins, *ipt*-transgenic plants, *Nicotiana tabacum*, water relations

## Abstract

Cytokinins are known to keep stomata open, which supports gas exchange and correlates with increased photosynthesis. However, keeping the stomata open can be detrimental if the increased transpiration is not compensated for by water supply to the shoots. In this study, we traced the effect of *ipt* (isopentenyl transferase) gene induction, which increases the concentration of cytokinins in transgenic tobacco plants, on transpiration and hydraulic conductivity. Since water flow depends on the conductivity of the apoplast, the deposition of lignin and suberin in the apoplast was studied by staining with berberine. The effect of an increased concentration of cytokinins on the flow of water through aquaporins (AQPs) was revealed by inhibition of AQPs with HgCl_2_. It was shown that an elevated concentration of cytokinins in *ipt*-transgenic plants increases hydraulic conductivity by enhancing the activity of aquaporins and reducing the formation of apoplastic barriers. The simultaneous effect of cytokinins on both stomatal and hydraulic conductivity makes it possible to coordinate the evaporation of water from leaves and its flow from roots to leaves, thereby maintaining the water balance and leaf hydration.

## 1. Introduction

Cytokinins promote the growth of plants, maximizing their productivity [1]. A decrease in the content of active cytokinins in stressed plants is accompanied by growth inhibition, and optimal mineral nutrition promotes the biosynthesis of cytokinins, thereby activating plant growth [2]. The mechanisms of action of cytokinins are well documented, indicating their influences on numerous processes in plants, one of which is the ability to keep stomata open [3,4,5,6]. This property of cytokinins is very important because high stomatal conductance supports gas exchange and correlates with increased photosynthesis, while stomatal closure, although conserving water by reducing transpiration, limits photosynthesis, resulting in decreased biomass accumulation and lowered yield [7]. Although the mechanisms of action of cytokinins are less understood than those of their antagonist abscisic acid (ABA), which is known to close stomata, there is some evidence to suggest how cytokinins act on stomata. The opening of stomata by cytokinins has been linked to their ability to reduce levels of hydrogen peroxide, which is known to be involved in stomatal closure [8]. In the *ahk2-2 ahk3-3* (Arabidopsis histidine kinases) double mutant of Arabidopsis with impaired cytokinin perception, the stomata did not open in the light to the same extent as in wild-type (WT) plants [9], while mutants with defective cytokinin type-B response regulators had lower stomatal conductance [10]. In contrast, upregulation of the expression of cytokinin-signaling genes prepared the grapevine for subsequent salt stress, helping to better keep stomata open and thereby increasing carbon assimilation [11].

Although the action of cytokinins on stomata appears to be beneficial for plants, keeping stomata open can be detrimental unless increased transpiration is compensated for by water supply to the shoots. The increase in stomatal conductance and transpiration must be balanced by an increase in plant hydraulic conductance, minimizing changes in leaf water potential. In this regard, the roots are of particular importance, since they have high hydraulic resistance, which limits water flow from the soil to the shoots [12]. Higher root hydraulic conductance was shown to be associated with water stress tolerance by sustaining a less negative leaf water potential and delaying stomatal closure [13]. Little attention has been paid to the role of cytokinins in the regulation of root hydraulic conductance. The review by Maurel and Nacry only mentions the ability of cytokinins to influence root growth and development, while considering the convergence of root hydraulics with root architecture [14]. However, cytokinins are known to inhibit root growth and branching [15], and such effects are unlikely to counterbalance the cytokinin-induced increase in transpiration. Otherwise, cytokinins may be involved in the control of root hydraulic conductivity (water flow rate related to root surface area and driving force), which is known to depend on transcellular transport through aquaporins (AQPs) and the formation of apoplastic barriers [16]. Information about the effects of cytokinins on these parameters is rather scarce and mostly indirect. A characterization of the rice cytokinin-responsive transcriptome revealed an upregulation of four aquaporin genes [17], while the inhibition of cytokinin signaling affects aquaporin phosphorylation [18]. A review on the hormonal control of secondary wall formation notes that the role of cytokinins in the control of genes involved in the biosynthesis of the secondary root wall remains unclear. However, the authors draw conclusions from endothecial lignification studies and report that plants overexpressing *AHP4*, a member of the multi-step phosphorelay signaling pathway and a positive regulator of cytokinin signaling, showed reduced endothecial lignification, while the *ahp4*-deficient mutant showed greater lignification compared to the WT plants. Cytokinins have also been shown to control the deposition of suberin in Casparian bands [19].

However, we were unable to find information on any effect of cytokinins on the hydraulic conductivity of the roots, and the present study aimed to fill the gap in this knowledge. We chose model transgenic tobacco plants transformed with the bacterial cytokinin biosynthesis gene isopentenyltransferase (*ipt*) cloned behind the heatshock 70 promoter from *Drosophila melanogaster* in order to trace the effect of increased cytokinin concentration on the hydraulic conductivity of the plants. This model has been previously used for the study of cytokinin effects on stomatal conductance [5]. In the present study, we traced the involvement of aquaporins in cytokinin-induced changes in hydraulic conductivity by inhibiting their activity with mercuric chloride. Mercurial compounds were found to reduce water transport in the cell membrane, and this selective inhibition subsequently allowed for aquaporin isolation and membrane transport characterization [20]. The formation of apoplastic barriers was studied by staining lignin and suberin with berberine. Thus, the aim of this study was to reveal the effects of cytokinins on hydraulic conductivity brought about by the formation of apoplastic barriers and changes in the activity of aquaporins using the model transgenic plants of tobacco, in which the induction of the *ipt* gene by heat shock increased the concentration of cytokinins.

## 2. Results

### 2.1. Concentration and Immunolocalization of Hormones

The comparison of cytokinin concentrations in both the roots or leaves of WT and transgenic (*IPT*) plants grown without heating did not reveal a significant difference between them (Figure 1). As expected, induction of the *ipt* gene by heat-shock (HS) treatment increased cytokinin concentration in both the leaves and roots of transgenic plants (Figure 1) detected 1 h after the HS treatment, while it did not significantly change the content of these hormones in WT plants. Cytokinin concentration in HS-treated transgenic plants decreased over time, but was increased again with the next HS treatment.

ABA concentration was similar in the leaves and roots of WT and transgenic plants grown without heating and its concentration in the leaves of WT plants was not affected by HS treatment, but ABA decreased in the leaves of HS-treated transgenic plants compared to the control *IPT* plants (Figure 2). ABA concentration in the roots increased upon treatment of both genotypes.

Under constant temperature, the IAA concentration in both the roots and leaves of WT plants was similar to that of the control transgenic plants and was not affected by the treatment, while root IAA concentration in transgenic plants was increased by the HS (Figure 3).

The immunohistochemical examination of sections of the basal part of tap (Figure 4) and lateral (Figure 5) tobacco roots using antibodies with high immunoreactivity to zeatin showed an increase in fluorescence after HS treatment, confirming an increase in the concentration of cytokinin in the roots (Figure 1).

### 2.2. Hydraulic Conductivity

The calculation of the hydraulic conductivity between the nutrient solution and the leaves confirmed that, after the first and third HS treatments, it was higher in *IPT* plants compared to WT plants (Figure 6).

To test the possible involvement of AQP in increasing the hydraulic conductivity of HS-treated transgenic plants, we treated them with mercuric chloride (AQP inhibitor). In our previous experiments [21], where HgCl_2_ was applied to plants in the same way as in the present experiments, its inhibitory effect was reversed by dithiothreitol, which is a sulfhydryl-reducing agent and heavy metal chelator. The reversal of the inhibitory effect of HgCl_2_ by dithiothreitol confirmed that such a short-term exposure, as in the present experiments, does not lead to an irreversible toxic effect. The calculation of hydraulic conductivity showed its decrease by four times in transgenic plants and only by two times in wild-type plants as a result of HgCl_2_ treatment (Figure 7). The results suggest a greater contribution of AQPs to the hydraulic conductivity of HS-treated *IPT* plants compared to WT plants.

After several days of HS treatment, its effect on the formation of apoplastic barriers may be expected, so we studied the deposition of lignin and suberin in the roots by staining their sections with berberine. Figure 8 shows that the fluorescence of this dye was slightly increased by HS treatment in the endoderm of both wild-type and transgenic plants, although the deposition of lignin and suberin in the endoderm was not high, as indicated by green coding, corresponding to low fluorescence according to the heat map. The fluorescence of the dye was significantly increased by HS treatment in the exodermis of WT plants, where it was encoded with blue and red corresponding to higher levels of lignin and suberin deposition. Fluorescence was lower in the exodermis of lateral roots of HS-treated transgenic plants (no blue or red and only green colors), suggesting that the induction of the *ipt* gene by the treatment decreased the formation of apoplastic barriers in the roots that were formed after the start of HS treatment.

To confirm importance of AQPs for the detected changes in hydraulic conductance, we measured the expression of AQP genes in the plants.

The highest level of the transcripts was detected for the *NtPIP2;1* gene (Figure 9). *PIP2* expression measured after the first HS did not differ between transgenic and wild tobacco plants (Figure 9A). The pattern changed after the third HS and higher levels of *NtPIP2;1*, *NtPIP2;4*, and *NtPIP2;5* transcripts were found in *IPT* than in wild type (Figure 9B).

## 3. Discussion

The increase in cytokinin content in the leaves of transgenic plants (Figure 1) caused by HS induction of the *ipt* gene was accompanied by an acceleration of transpiration (Figure 6). These results are in accordance with data showing the ability of cytokinins to keep stomata open [3,4]. We have previously shown that induction of the *ipt* gene in the same transgenic plants led to an increase in stomatal conductance and transpiration [5]. The novelty of the present research lies in our attempt to study the consequences of increased transpiration in these transgenic plants for their water relations and their dependence on changes in hydraulic conductance. The measurement of hydraulic conductivity (Figure 7) showed that it was higher in HS-treated transgenic plants. Hydraulic conductance is known to increase in accordance with the evaporation demand [22]. However, the mechanism of this effect is still unclear, so it is important to find out how it was realized in the present experiments.

Experiments with the AQP inhibitor and the measurement of *HvPIP2* gene expression have shown the involvement of these water channels in the increase in hydraulic conductivity in HS-treated transgenic plants characterized by increased cytokinin content. Information on the ability of cytokinins to influence AQPs’ activity is scarce. Therefore, we attempted to relate the changes in hydraulic conductivity found in the present experiments to HS-induced changes in other hormones.

It is well known that ABA affects the activity of AQPs [23], and increased hydraulic conductivity could be associated with the accumulation of this hormone found in the roots of transgenic plants treated with HS. However, ABA content in the roots was increased upon treatment with HS in both transgenic and WT plants (Figure 2), while the increase in hydraulic conductivity was typical only for transgenic plants, which argues against the importance of ABA for the control of hydraulic conductivity in the present experiments.

In contrast to ABA, the HS-induced increase in the content of IAA in the roots was found only in *IPT* plants (Figure 3). However, it was not possible to associate changes in auxins with an increase in the hydraulic conductivity of plants, since it is known that auxins decrease rather than increase the activity of AQPs [24].

In the present experiments, we found increased abundance of three out of four *PIP2* gene transcripts in transgenic plants after the third HS, which confirmed the involvement of AQP gene expression in the control of cytokinin-induced changes in hydraulic conductivity. Available transcriptomic data indicate the ability of cytokinins to increase the expression of genes encoding AQPs [17]. However, the HS-induced changes in hydraulic conductivity found in transgenic plants after the first heat shock were not associated with changes in gene expression. It is more likely that changes in hydraulic conductivity were associated with AQP phosphorylation, since cytokinins are known to influence this process [18]. Although the significance of cytokinins in controlling water flow through plants needs further testing, the data obtained in the present experiments support the possibility of their involvement in the control of hydraulic conductivity through their influence on AQP activity.

Repeated treatment with HS for several days affected the formation of apoplastic barriers in new lateral roots of transgenic plants (Figure 9). Heat-shock factors have been associated with lignin biosynthesis [25], which may explain the slightly elevated fluorescence of berberin found in the endodermis of both WT and transgenic HS-treated plants. Increased formation of apoplast barriers was more highly pronounced in the exodermis of WT plants, while no such effect was detected in ipt-transgenic plants. Since the formation of apoplastic barriers decreases hydraulic conductivity [26], the reduced deposition of lignin and suberin found in the exodermis of HS-treated transgenic plants may contribute to increased plant hydraulic conductivity. Again, information on the involvement of cytokinins in the control of the process is scarce. Data on ABA are more numerous, but this hormone increases rather than decreases the deposition of suberin [27,28,29], while the accumulation of this hormone in the roots of HS-treated transgenic plants was accompanied by a decrease in the staining of lignin and suberin with berberine. The same argument excludes the possibility of auxin’s participation in the control of the process. Although a recent report has shown the need for auxin-mediated transcriptional responses to modify suberin synthesis [30], its increased concentration in the roots of HS-treated transgenic plants can hardly be the reason for the weaker development of apoplast barriers in the lateral roots found in the present experiments.

Therefore, we again had to turn to cytokinins in search of the mechanism responsible for the regulation of exodermis formation in lateral roots. Although the role of cytokinins in the control of genes involved in the biosynthesis of root secondary walls remains unclear [31], some experiments have shown the involvement of cytokinins in reducing the lignification of the endothecium [32]. Arabidopsis histidine-containing phosphotransfer factor 4 (AHP4) negatively regulates the secondary wall thickening of the anther endothecium wall during flowering and deposition of suberin in Casparian bands [19]. This information, combined with our data on the weaker deposition of lignin and suberin in the exodermis of lateral roots of HS-treated transgenic plants, suggests that an increased level of cytokinin in plants is involved in the control of the process.

## 4. Materials and Methods

Experiments were carried out on wild (WT) and transgenic (HS*IPT*) tobacco plants (*Nicotiana tabacum* L. cv. Petit Havana SR-1).

### 4.1. Experimental Design

Seeds of wild (WT) and transgenic tobacco (HS*IPT*) were obtained as described previously [33]. Plants of the transgenic line were transformed with the bacterial ipt gene responsible for the synthesis of isopentenyl transferase cloned behind the heat shock (HS) protein promoter of *Drosophila melanogaster*. In our experiments, the seeds were germinated in the soil for 7 days, and the seedlings were planted in 300 cm^3^ pots with sand saturated with 100% Hoagland–Arnon nutrient mixture (0.5 mM KNO_3_, 0.5 mM Ca(NO_3_)_2_, 0.1 mM KH_2_PO_4_, 0.1 mM MgSO_4_ with essential microelements) and grown at a 16 h photoperiod, illumination of 400 µmol m^−2^s^−1^ PAR, and a temperature of 26/18 °C (day/night). Plants were supplied with mineral nutrients by daily watering with 5 mL of 100% Hoagland–Arnon solution and distilled water was added to 70% of the total water capacity of the sand. Plants at the stage of six leaves, excluding cotyledons, were subjected to total heating (HS) in a thermostat for an hour at a temperature of 40–42 °C. Control plants were placed in a closed box at room temperature. After the treatments, the plants were transferred to growing conditions. HS treatment was repeated 3 times (the second HS one day after the first and the third HS on the fourth day). Sampling for PCR and cytokinin analysis was performed after the first and the third HSs. Hydraulic conductivity was assayed as described below simultaneously with sampling for PCR and cytokinin analysis. Exact timing of sampling is indicated in figure legends. Sections of the basal part of the lateral roots were prepared after the third HS treatment for staining with berberine. To inhibit aquaporins, HgCl_2_ at concentration of 0.5 mM was added to the root environment before heating the plants.

### 4.2. Parameters of Water Relations

#### 4.2.1. Transpiration

Transpiration was assessed as water loss measured by weighing the pots. The surface of the sand was covered with aluminum foil to prevent water from evaporating directly from the surface of the sand.

#### 4.2.2. Hydraulic Conductivity

The hydraulic conductivity of the water transport pathway from roots to leaves was calculated using the formula L = T/(Ψs − Ψl) m, where T is transpiration, Ψs and Ψl are the water potential of the nutrient solution and leaf, respectively, and m is root mass (modification of method [34]). Water potential of leaf was measured with a psychrometer PSYPRO (Wescor, Logan, UT, USA). Water potential of the nutrient solution was measured with osmometer Osmomat 030 (Gonotec GmbH, Berlin, Germany).

### 4.3. Hormone Analyses and Immunolocalization

#### 4.3.1. Hormone Analyses

To extract hormones from leaves and roots, they were homogenized in 80% ethanol and incubated overnight at 4 °C. After filtration and evaporation of ethanol, the aqueous residue was divided into parts, one part was concentrated on a C18 column (Bond-Elut, RP-C18) and zeatin (Z), zeatin riboside (ZR), and zeatin nucleotide (ZN) were separated using TLC for their determination by enzyme immunoassay (ELISA) as described [5]. IAA and ABA were extracted from the other part with dimethyl ether [35]. IAA and ABA were extracted twice from the aqueous residue acidified to pH 2.5, with diethyl ether at a ratio of the organic and aqueous phases of 1:3. An amount of 1% NaHCO_3_ solution was added to the combined organic fraction (organic/aqueous phase ratio was 3:1), shaken, and ether was discarded. After acidification of the aqueous solution containing hormones, ether re-extraction was carried out by reducing the volume of the extractant (3:1). Reducing the amount of extractant at each stage of extraction and re-extraction increased selectivity of hormone recovery [36]. Hormone methylation was performed by adding diethyl ether containing diazomethane. After evaporation of the diethyl ether, the hormones were dissolved in 80% ethanol and aliquots were taken for quantification by ELISA using specific antibodies as described [37].

#### 4.3.2. Immunohistochemistry

Immunolocalization of cytokinins was carried out in basal part of tap and lateral roots from *ipt*-transformed and control (WT) tobacco plants as described [37,38]. Three-millimeter-long tissue pieces were collected and fixed in the mixture of 4% paraformaldehyde (Riedel de Haen, Seelze, Germany) and 0.1% glutaradehyde (Sigma, Steinheim, Germany) in PBS (pH 7.4). To improve the penetration of the fixative into the plant tissues, the initial stage (first 30 min) of fixation was carried out under vacuum. After infiltration, plant pieces were transferred to a fresh fixative and kept at 4 C overnight. After dehydration, an ascending series of ethanol plant tissues was embedded in JB4 resin (Electron Microscopy Sciences, Hatfield, PA, USA). Histological sections 1.5 µm thick were obtained on a microtome HM 325 MICROM (Laborgerate, Walldorf, Germany) and placed on glass slides. Before incubating overnight (at 4 °C) with the anti-Cytokinin (1:100) polyclonal rabbit antibody, samples were pre-treated with PBS containing 0.2% gelatin and 0.05% Tween-20 (PGT, which was used for antibody dilution) for 30 min to reduce non-specific binding. After washing three times with 0.05% Tween-20 in PBS for 10 min, sections were incubated with anti-rabbit IgG secondary antibodies conjugated to Alexa Fluor 555 (Invitrogen, Rockford, IL, USA) (1:500 in PGT) for 3 h at 37 °C in darkness. Samples were washed five times for 10 min with PBS and rinsed in MilliQ water. Then, the sections were coated with a mixture of glycerin and gelatin (0.5 g of gelatin, 3.5 mL of glycerin in 3 mL of water), were immediately covered by cover slips, and were viewed with a confocal laser scanning microscope using an FV3000 FluoView (FV31-HSD) (Olympus, Tokyo, Japan) and laser excitation lines of 561 nm.

#### 4.3.3. Lignin and Suberin Staining with Berberine Hemisulfate

Lignin and suberin were identified as described in [39]. Basal pieces of lateral root were fixed in a mixture (7:7:100) of formalin, acetic acid, and ethanol 70%, dehydrated in an ethanol series, cleared with xylene, and embedded in paraffin. Sections were cut on a microtome and, after a dewaxing step, were stained using an aqueous solution of berberine hemisulfate (0.1% *w*/*v*) for 1 h and then rinsed 2 times with distilled water. To enhance the fluorescence intensity, sections were additionally stained for 15 min with toluidine blue (0.05% *w*/*v*) in 0.1 M phosphate buffer (pH 5.6), rinsed 2 times with distilled water, mounted in a 0.1% FeCl_3_/50% glycerol mixture, and covered with a cover slip. Sections were excited with a 488 nm solid-state laser using a confocal laser scanning confocal microscope Olympus FluoView FV3000 (Olympus, Tokyo, Japan). Fluorescence emission was detected at 520 nm.

### 4.4. RNA Extraction and Analysis of Abundance of PIP2 mRNA

RNA was extracted using TRIzol ™ Reagent (Sigma, Steinheim, Germany) according to the manufacturer’s instructions. Potential contaminating DNA was digested with DNaseI (Synthol, Moscow, Russia) and first-strand cDNA was synthesized using the M-MLV reverse transcriptase (Fermentas, Waltham, MA, USA). Oligo(dT)15 was used as a primer, and the reverse-transcription reagents were incubated at 37◦C for 1 h in a total volume of 25 μL. After ten-fold dilution, 2 μL of the synthesized cDNA was used for quantitative real-time polymerase chain reaction (qPCR). The primers for qPCR were designed based on the cDNA sequence [40] using the PrimerQuest™ Tool (standard version). The primers used for quantitative analysis of *NtPIP2;1*, *NtPIP2;4*, *NtPIP2;5*, and *NtPIP2;6* are given in Table 1. *PIP2* genes were selected for analysis on the base of the data [40] indicating their high abundance in tobacco root.

Quantitative PCR was performed by polymerase chain reaction in real time using a set of predefined reagents, EvaGreenI (Synthol, Moscow, Russia), and QuantStudio™5 Real-Time PCR System by Thermo Fisher Scientific (Applied Biosystems, Waltham, MA, USA). The qPCR program was as follows: 95 °C for 5 min; 40 cycles of 95 °C for 15 s, 60 °C for 20 s, and 72 °C for 30 s. After the final PCR cycle, a melting curve analysis was conducted to determine the specificity of the reaction (at 95 °C for 15 s, 60 °C for 1 min and 95 °C for 15 s). The efficiency of each primer pair was determined using a 10-fold cDNA dilution series to reliably determine the fold changes. The ribosomal protein gene (NtL25) [41] was chosen as an internal control to normalize the amount of total RNA present in each reaction (Table 1).

All reactions, including the non-template control, were performed three times. The threshold values (CT), generated from the CFX Connect real-time PCR Detection System software tool (version 2.2) (Applied Biosystems, Waltham, MA, USA), were employed to quantify relative gene expression using the comparative threshold (delta CT) method. Three independent biological replicates were performed for each experimental variant.

### 4.5. Statistics

The data were statistically processed using Statistica version 10 software (Statsoft, Moscow, Russia). The tables and figures show means and their standard errors. The significance of differences was assessed by ANOVA followed by Duncan’s *t*-test (*p* ≤ 0.05).

## 5. Conclusions

Experiments on transgenic tobacco plants with HS-induced accumulation of cytokinins showed that an elevated concentration of cytokinins not only increases transpiration, which was shown earlier, but also upregulates hydraulic conductivity, which, as far as we know, was shown by us for the first time. This effect was due to the cytokinin-induced increase in expression of AQP genes and their activity and decreased formation of apoplast barriers. The simultaneous effects of cytokinins on both stomatal and hydraulic conductivity make it possible to coordinate the evaporation of water from leaves and its flow from roots to leaves, thereby maintaining water balance and leaf hydration.

## Figures and Tables

**Figure 1 ijms-24-09860-f001:**
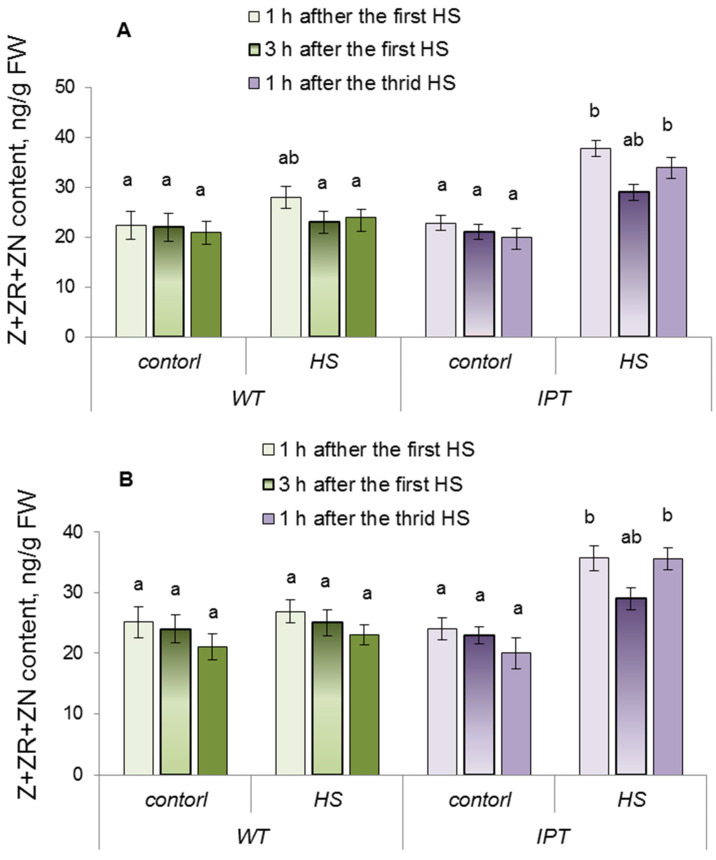
Total cytokinin content (sum of zeatin (Z), zeatin riboside (ZR), and zeatin nucleotide (ZN) per fresh weight (FW)) in the third and fourth leaves (**A**) and roots (**B**) of wild-type (WT) and transgenic (*IPT*) tobacco plants 1 and 3 h after the first HS treatment and 1 h after the third HS at 40–42 °C. Significantly different means are marked with different letters; *p* < 0.05, *n* = 9. ANOVA followed by Duncan’s test at *p* < 0.05, *n* = 9.

**Figure 2 ijms-24-09860-f002:**
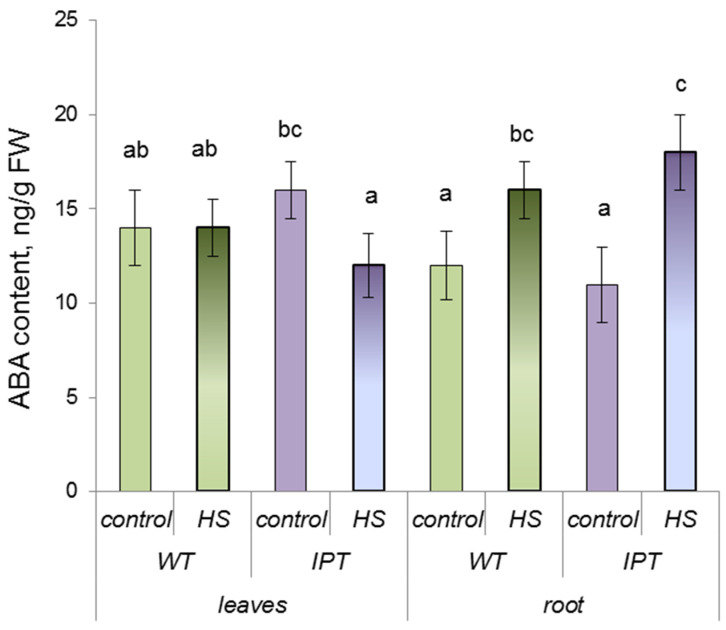
ABA content (per FW) in the third and fourth leaves and roots of wild-type (WT) and transgenic (*IPT*) tobacco plants 1 h after the first HS treatment at 40–42 °C. Significantly different means are labeled with different letters (ANOVA followed by Duncan’s test at *p* < 0.05, *n* = 9).

**Figure 3 ijms-24-09860-f003:**
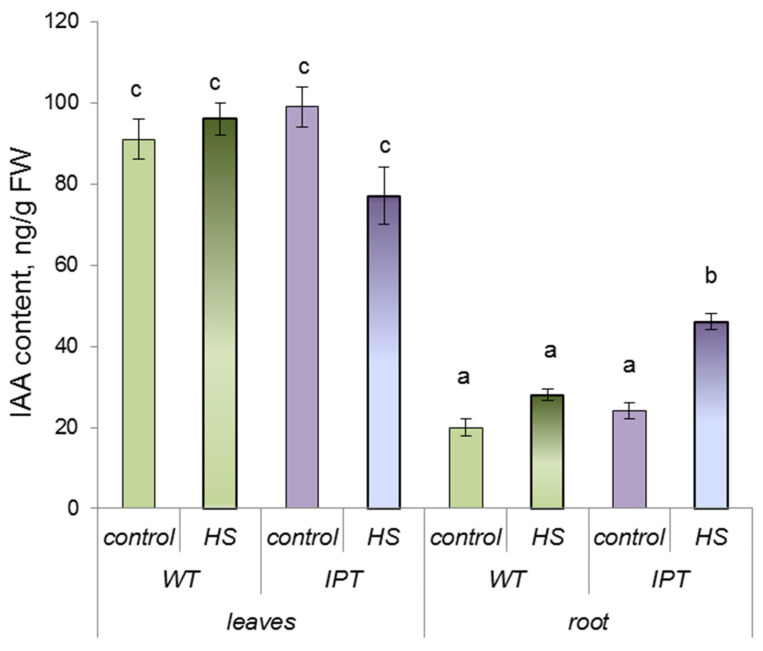
IAA content (per FW) in the third and fourth leaves and roots of wild-type (WT) and transgenic (*IPT*) tobacco plants 1 h after the first HS treatment at 40–42 °C. Significantly different means are labeled with different letters (ANOVA followed by Duncan’s test at *p* < 0.05, *n* = 9).

**Figure 4 ijms-24-09860-f004:**
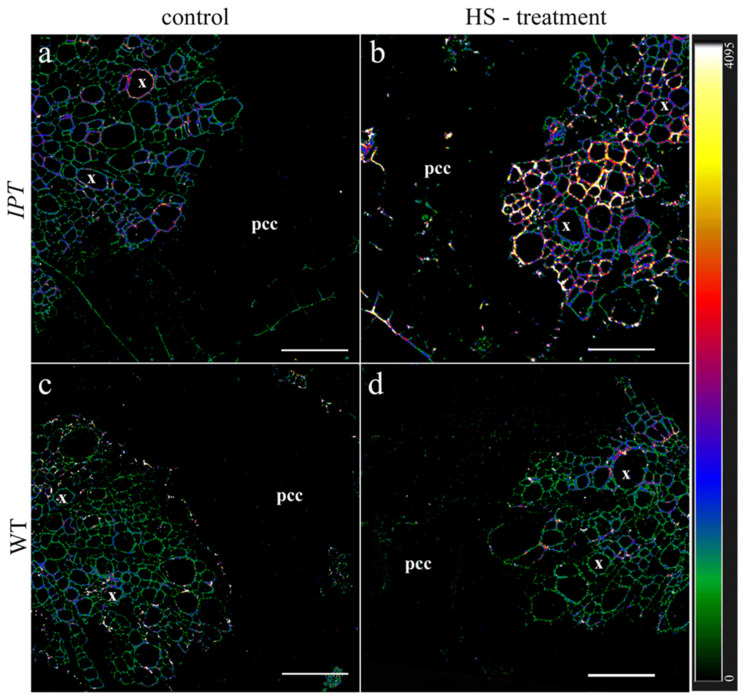
Immunolocalization of cytokinins in central cylinder of the tap roots of transgeninc (*IPT*) (**a**,**b**) and wild-type (WT) (**c**,**d**) tobacco plants grown without heating (control) (**a**,**c**) or 1 h after heat shock (HS) treatment (**b**,**d**). x—xylem, pcc—parenchyma of central cylinder. Heatmap represents intensity of fluorescence corresponding to cytokinins depicted by color. Scale bar is 100 µm.

**Figure 5 ijms-24-09860-f005:**
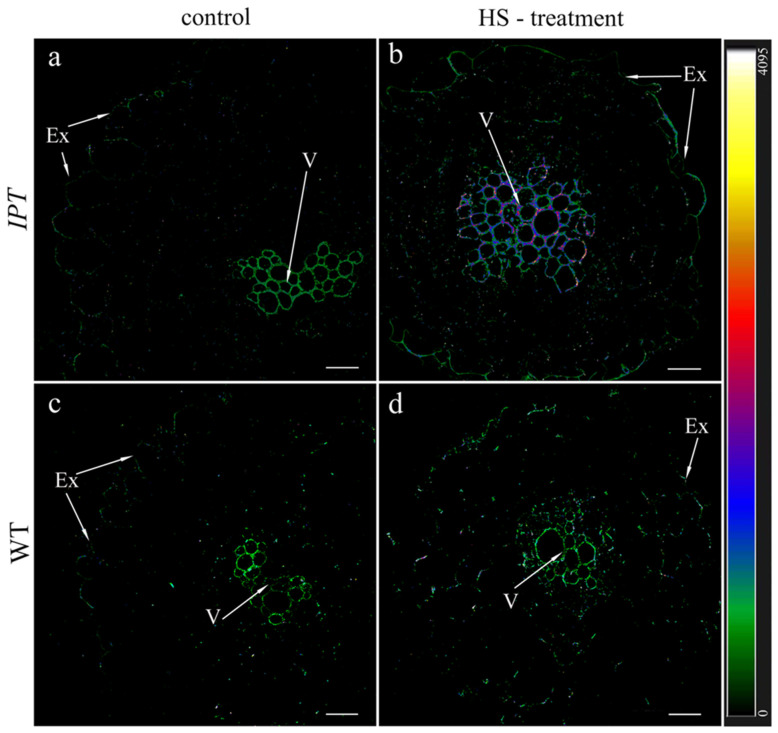
Immunolocalization of cytokinins in root cross sections of basal part of lateral roots of transgenic (*IPT*) (**a**,**b**) and wild-type (WT) (**c**,**d**) tobacco plants grown without heating (control) (**a**,**c**) or 1 h after heat shock (HS) treatment (**b**,**d**). Ex—exodermis, V—vasculature. Heatmap represents intensity of fluorescence corresponding to cytokinins depicted by color. Scale bar is 50 µm.

**Figure 6 ijms-24-09860-f006:**
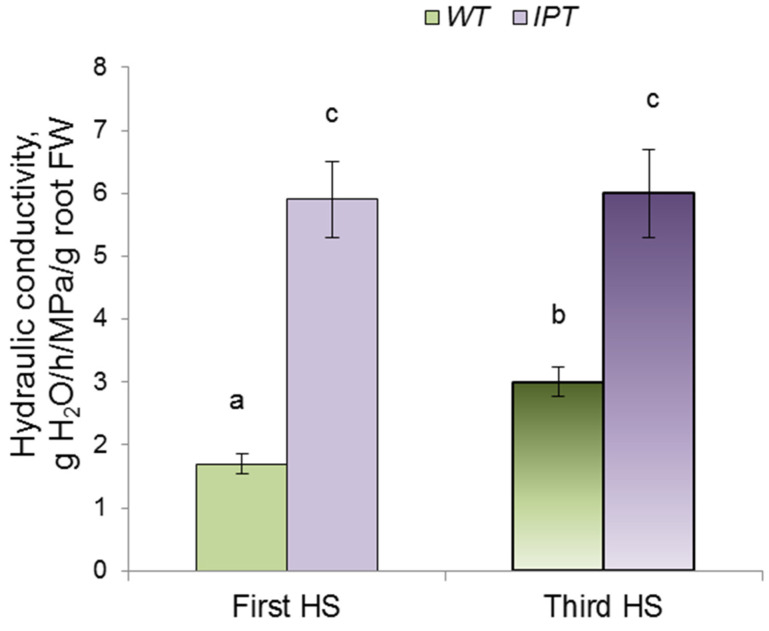
Hydraulic conductivity of wild-type (WT) and transgenic (*IPT*) tobacco plants measured after the first and third HS treatment at 40–42 °C. Significantly different means are labeled with different letters (ANOVA followed by Duncan’s test at *p* < 0.05, *n* = 9).

**Figure 7 ijms-24-09860-f007:**
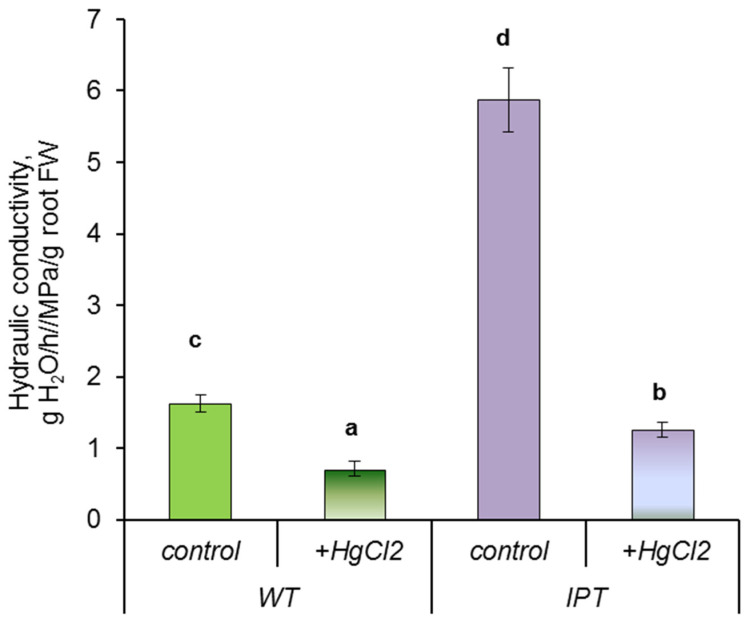
Hydraulic conductivity of control and pretreated with 0.5 mM HgCl_2_ wild-type (WT) and transgenic (*IPT*) tobacco plants 1 h after HS treatment at 40–42 °C. Significantly different means are labeled with different letters (ANOVA followed by Duncan’s test at *p* < 0.05, *n* =8).

**Figure 8 ijms-24-09860-f008:**
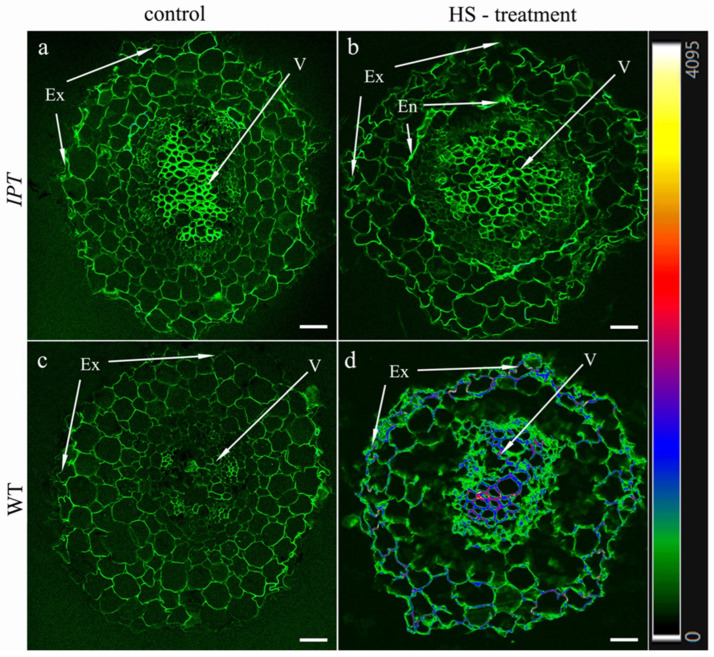
Localization of lignin and suberin in berberine-stained root cross sections of transgenic (*IPT*) (**a**,**b**) and wild-type (WT) (**c**,**d**) control (unheated) (**a**,**c**) tobacco plants and those HS-treated thrice over 4 days (**b**,**d**): En—endodermis, Ex—exodermis, V—vasculature. Heatmap represents color-coded intensity of fluorescence corresponding to lignin and suberin. Scale bar: 50 µm.

**Figure 9 ijms-24-09860-f009:**
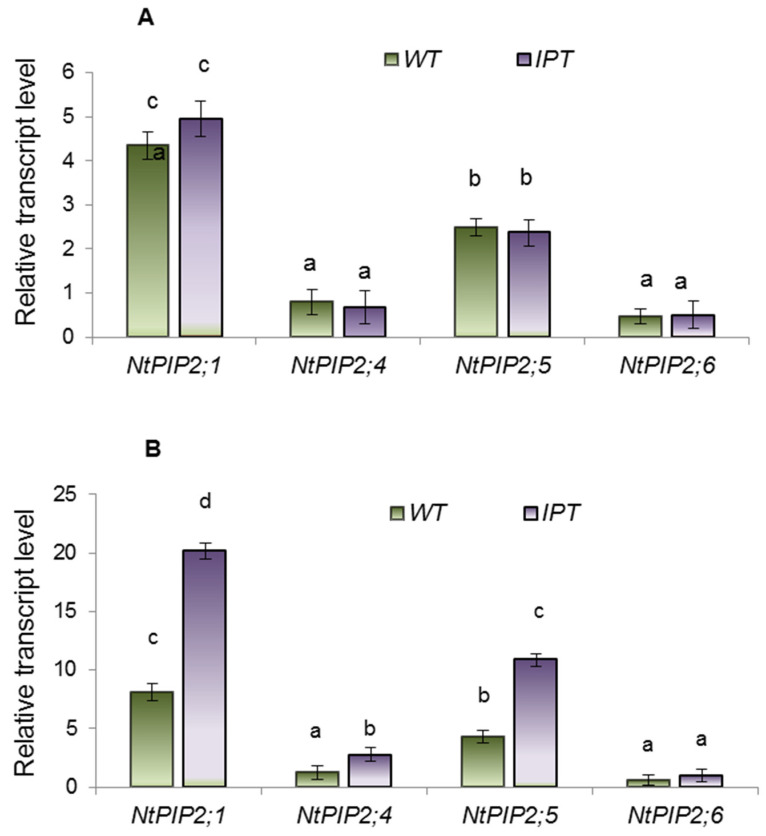
Transcript level of PIP2 AQP genes of the wild-type (WT) and transgenic (*IPT*) tobacco plants measured after the first (**A**) and third (**B**) HS treatments. Significantly different means are labeled with different letters (ANOVA followed by Duncan’s test at *p* < 0.05, *n* = 9).

**Table 1 ijms-24-09860-t001:** Sequences of primers used for qRT-PCR.

Genes	Strand	5′ to 3′ Primer Sequences	GenBank Accession Number
*NtPIP2;1*	Forward	AACAGCACGGGAAGGATTAC	AF440272.1
	Reverse	GAGTAGCAATGAACTCAGCAATAAG	
*NtPIP2;4*	Forward	GGTATGGTGGAGGTGCTAATG	BK011406
	Reverse	GTGGCAGAGAAGACAGTGTAAA	
*NtPIP2;5*	Forward	GGACATATTAACCCAGCAGTGA	BK011408
	Reverse	CAACCACAAATGGCTCCTAAAC	
*NtPIP2;6*	Forward	TTCTCTGCTACTGACCCTAAGA	BK011410
	Reverse	TGGCAAGGTGAACCATGAATA	
*NtL25*	Forward	GGCTGTCAAGTCAGGATCAA	L18908.1
	Reverse	GTTCCTTCCAGGTGCACTAATA	

## Data Availability

The data presented in this study are available in the graphs and tables provided in the manuscript.

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
