# Peer review of "Effect of ipt Gene Induction in Transgenic Tobacco Plants on Hydraulic Conductance, Formation of Apoplastic Barriers and Aquaporin Activity under Heat Shock"

_ijms, 2023, doi:10.3390/ijms24129860_

Round 1

Reviewer 1 Report

In this study, the effect of induction of the ipt gene, which increases cytokinin concentration in transgenic tobacco plants, on transpiration and hydraulic conductance was investigated. The role of aquaporins was evaluated by inhibition with HgCl2. And it was shown that increasing cytokinin concentrations in ipt-transgenic plants increased hydraulic conductance by enhancing aquaporin activity and reducing apoplastic barrier formation. 

I had a few questions after reading the paper. 

1) So, lines 95-96 describe the dynamics with a decrease in cytokinin concentration after shock, but these results are not shown. 

2) What is the criterion for the specific effect of HgCl2 in the experiment on aquaporins? How was general toxicity excluded?

3) According to Figure 9 in transgenic tobacco, the control and shock differ in the presence of stronger staining of the endoderm of the shocked plant, which can be seen as a ring.  This is not discussed in the article, but draws attention. How regular was this result? And can this image characterize the effect of heat shock on the formation of barriers in the apoplast?

Author Response

We are most grateful to the reviewer for attentive reading of our article and valuable comments, which we carefully followed

1. lines 95-96 describe the dynamics with a decrease in cytokinin concentration after shock, but these results are not shown.

Response: We added the data on dynamics of cytokinins 

2) What is the criterion for the specific effect of HgCl2 in the experiment on aquaporins? How was general toxicity excluded?

Response: To justify the use of HgCl2  we added to the Introduction that “Mercurial compounds were found to reduce water transport in the cell membrane and this selective inhibition subsequently allowed for aquaporin isolation and membrane transport characterization [20].” And to Result section that “The effect of HgCl2 was apparently non-toxic, which was confirmed in our previous experiments showing that dithiothreitol treatment reversed the inhibitory effect of HgCl2 on water relations in such short-term experiments [21].”

3) According to Figure 8 in transgenic tobacco, the control and shock differ in the presence of stronger staining of the endoderm of the shocked plant, which can be seen as a ring.  This is not discussed in the article, but draws attention. How regular was this result? And can this image characterize the effect of heat shock on the formation of barriers in the apoplast?

Response: In accordance with this remark we modified description of these results: “Figure 8 shows that the fluorescence of this dye was slightly increased by HS treatment in endoderm of both wild-type and transgenic plants, although deposition of lignin and suberin was not high in endodrem, as indicated by green coding, corresponding to low fluorescence. Fluorescence of the dye was significantly increased by HS-treatment in exodermis of WT plants, where it was encoded with blue and red corresponding to higher levels of lignin and suberin deposition. Fluorescence was lower in exodermis of lateral roots of HS-treated transgenic plants (no blue or red and only green colors) suggesting that induction of ipt gene by the treatment decreased formation of apoplastic barriers in the roots that were formed after the start of HS-treatment.” We also added to Discussion that “Heat shock factors have been associated with lignin biosynthesis [25], which may explain the slightly elevated fluorescence of berberin found in endodermis of both WT and transgenic HS-treated plants. Increased formation of apoplast barriers was much greater pronounced in exodermis of WT plants, while no such effect was detected in ipt-transgenic plants.”

Reviewer 2 Report

The manuscript by Vysotskaya et al. relates with the role of cytokinins in the responses of plants to heat stress, and their effect on water relations in plants. The manuscript lacks clear objectives and hypothesis, the design of the experiments is quite poor, with consequences in the results and the consistency of the discussion. My main concerns are:

1. Title: The title does not mention the main treatment of the study: heat stress.

2. The introduction needs to be improved, it is now well-written and it is difficult to follow. The manuscript lacks any objectives and hypothesis. 

3. The methodology used is not well-explained, specially how the heat stress treatment was applied and how the different determinations were made. It seems that heat stress was induced during different days, although the determinations presented over the paper are done at different time points. 

- Hormones, inmunolocalization, transpiration and hydraulic conductivity: 1 h after HS-treatment at 40-42C

- Lignin and suberin after the series of HS-treatment for 4 days. Not only the timing is diffeent, but the root hydraulic conductivity is not meassured, so it is difficult to determine their role in whole plant water balance, there could have been other factors affecting it, not just the lignin and suberin accumulated in some sections of the root.

4. Mercury chloride is used for blocking the aquaporins. The authors shoulds have demonstrated that there is not a toxic effect of this product used with heat stress exposure. Further, the data should be supported with some other determinations, as expression of aquaporin genes and protein abundance.

Author Response

We are grateful to the reviewer for valuable critical comments. We have done our best to follow them. Thus additional experiment was performed on measurement of expression of AQP genes according to recommendation of reviewer, which took several months. The list of the changes made according to the recommendations of reviewer is as follows:

  1. Title: The title does not mention the main treatment of the study: heat stress.

Response: In accordance with the remark of reviewer, heat shock was mentioned in the title: “Effect of heat shock induction of ipt gene on hydraulic conductance, formation of apoplast barriers and aquaporins activity of transgenic tobacco plants“

  1. The introduction needs to be improved, it is now well-written and it is difficult to follow. The manuscript lacks any objectives and hypothesis.

Response:  To improve Introduction we added some information, which was obviously not clear enough. Thus we explained how ipt gene was induced in transgenic plants: “We chose model transgenic tobacco plants transformed with the bacterial cytokinin biosynthesis gene isopentenyltransferase (ipt) cloned behind the heatshock 70 promoter from Drosophila melanogaster in order to trace the effect of increased cytokinin concentration on the hydraulic conductivity of the plants.” We also added some explanations concerning the use of mercuric chloride:  “Mercurial compounds were found to reduce water transport in the cell membrane and this selective inhibition subsequently allowed for aquaporin isolation and membrane transport characterization [19]”. And at last, but not least, we tried to highlight the goal of the present research by telling “Thus, the aim of this study was to reveal the effects of cytokinins on hydraulic conductivity brought about by the changes in the activity of aquaporins and formation of apoplastic barriers using the model transgenic plants of tobacco, in which the induction of the ipt gene by heat shock increased concentration of cytokinins.”

  1. The methodology used is not well-explained, specially how the heat stress treatment was applied and how the different determinations were made. It seems that heat stress was induced during different days, although the determinations presented over the paper are done at different time points.

Response: We modified description of experimental design trying to clarify it and added some data obtained after repeated HS treatments. Description of HS treatment and sampling is as follows: “HS treatment was repeated 3 times (the second HS one day after the first and the third HS on the fourth day). Sampling for PCR and cytokinin analysis was performed after the first and the third HSs. Hydraulic conductivity was assayed as described below simultaneously with sampling for PCR and cytokinin analysis. Exact timing of sampling is indicated in the Figure legends. Sections of the basal part of the lateral roots were prepared after the third HS-treatment for staining with berberine. To inhibit aquaporins, HgCl2 at concentration of 0.5 mM was added to the root environment before heating the plants.”

  1. Mercury chloride is used for blocking the aquaporins. The authors shoulds have demonstrated that there is not a toxic effect of this product used with heat stress exposure. Further, the data should be supported with some other determinations, as expression of aquaporin genes and protein abundance.

Response: The effect of HgCl2 was apparently non-toxic, which was confirmed in our previous experiments showing that dithiothreitol treatment reversed the inhibitory effect of HgCl2 on water relations in such short-term experiments [20]. This information was added to the text.

Furthermore, we repeated experiments to measure expression of AQP genes. It took several months to grow tobacco plants and this is the reason of the delay in our submission of revised version. Description of RNA extraction and PCR were introduced into M & M section. In this section we justify the study of PIP2;1, PIP2;4, PIP2;5 and PIP2,6 genes by making a reference to the literature data showing their high abundance in tobacco roots [40]. We added to the Result section that “Highest level of the transcripts was detected for the NtPIP2;1 gene (Fig. 9). PIP2 expression measured after the first HS did not differ between transgenic and wild tobacco plants (Fig. 9a). The pattern changed after the third HS and higher level of NtPIP2;1, NtPIP2;4 and NtPIP2;5 transcripts was found in IPT than in wild type (Figure 9b.” To the discussion section we added that “In the present experiments we found increased abundance of 3 out of 4 PIP2 gene transcripts in transgenic plants after the third HS, which confirmed involvement of AQP genes expression in the control of cytokinin-induced changes in hydraulic conductivity.” We also modified one more sentence of discussion: “However, the HS-induced changes in hydraulic conductivity found in transgenic plants after the first heat shock were not associated with changes in gene expression”.

We also hope that Conclusion added to the revised version of the article might improve opinion of reviewer about it: “Experiments on transgenic tobacco plants with HS-induced accumulation of cytokinins showed that an elevated concentration of cytokinins not only increases tran-spiration, which was shown earlier, but also up-regulates hydraulic conductivity, which, as far as we know, was shown by us for the first time. This effect was due to cytokinin-induced increase in expression of AQP genes and their activity and decreased formation of apoplast barriers. The simultaneous effects of cytokinins on both stomatal and hydraulic conductivity make it possible to coordinate evaporation of water from leaves and its flow from roots to leaves, thereby maintaining water balance and leaf hydration.”

Reviewer 3 Report

Congratulations on your manuscript " Effect of induction of ipt gene on hydraulic conductance, formation of apoplast barriers and aquaporins activity of transgenic tobacco plants" (ijms-2217686). The introduction, materials and methods, results, discussion, and conclusion of your paper are very interesting, especially in regards to the relationship between ipt genes and hydraulic conductivity. Team group have done a remarkable amount of work and used various analytical methods and statistical analyses to support your findings. The experimental work was well executed and the manuscript presents the results in a logical and coherent manner, with most of the results backed up by experimental data. The structure of the manuscript is adequate, the information is new, and of great significance. The images are visually appealing and effectively convey the cause-and-effect relationship.

My minor suggestions are:

#1. Alphabetize the keywords.

#2. Please standardize the nomenclature of equipment/reagents/software by including the manufacturer, city, state, and country (using three-letter codes) when necessary. Please check the manuscript for consistency.

#4. Please consider updating old references, especially those published before 2010, if new references are available.

#5. Enhancing the conclusion section may further improve the visibility and impact of your manuscript for readers.

Best regards

Author Response

We are most grateful to the reviewer for high estimation of our article and valuable comments

#1. Alphabetize the keywords.

Response: We alphabetized the keywords

#2. Please standardize the nomenclature of equipment/reagents/software by including the manufacturer, city, state, and country (using three-letter codes) when necessary. Please check the manuscript for consistency.

Response: we standardized the nomenclature of equipment/reagents/software

#4. Please consider updating old references, especially those published before 2010, if new references are available.

Response: We are sorry for being unable to update some references, since involvement of cytokinins in the control of either stomatal or hydraulic conductance has not been frequently studied. Still recent references are dominating in our article. 28 cited articles (about two-thirds) have been published between 2013 and 2023 (15 of them in the last 5 years). Only 2 cited articles were published before 2000. One of them describes obtaining transgenic plants used in the present study (Schmulling et al., 1989) and the last is the original description of the modified method of hormone partitioning (Veselov et al., 1992).

#5. Enhancing the conclusion section may further improve the visibility and impact of your manuscript for readers.

Response: We added Conclusion, where it is said that ”Experiments on transgenic tobacco plants with HS-induced accumulation of cytokinins showed that an elevated concentration of cytokinins not only increases transpiration, which was shown earlier, but also up-regulates hydraulic conductivity, which, as far as we know, was shown by us for the first time. This effect was due to cytokinin-induced increase in expression of AQP genes and their activity and decreased formation of apoplast barriers. The simultaneous effects of cytokinins on both stomatal and hydraulic conductivity make it possible to coordinate evaporation of water from leaves and its flow from roots to leaves, thereby maintaining water balance and leaf hydration.”